# Lynch Syndrome: From Carcinogenesis to Prevention Interventions

**DOI:** 10.3390/cancers14174102

**Published:** 2022-08-24

**Authors:** Donatella Gambini, Stefano Ferrero, Elisabetta Kuhn

**Affiliations:** 1U.O.C. Oncologia Medica, Fondazione IRCCS Ca’ Granda Ospedale Maggiore Policlinico, 20122 Milano, Italy; 2U.O.C. Anatomia Patologica, Fondazione IRCCS Ca’ Granda Ospedale Maggiore Policlinico, 20122 Milano, Italy; 3Chirurgiche ed Odontoiatriche, Dipartimento di Scienze Biomediche, Università degli Studi di Milano, 20122 Milano, Italy

**Keywords:** Lynch syndrome, prevention, carcinogenesis, colorectal cancer, endometrial cancer, ovarian cancer

## Abstract

**Simple Summary:**

Promoting proper preventive interventions to reduce morbidity and mortality is one of the most important challenges pertaining to inherited conditions. Lynch syndrome (LS) is an inherited disorder that predisposes to several kinds of tumor and is responsible for a relevant proportion of human colorectal and endometrial cancers. Recent knowledge has allowed for a better understanding of the genetic cause, pathogenesis, underlying immunological mechanisms, epidemiological distribution, and prevalence of this disease. This opens up unpredictable perspectives of translating such knowledge into validated programs for prevention and surveillance, in order to reduce the health impact of this disease through medical interventions before cancer development. In our review, we summarize the updated guidelines of the screening, surveillance, and risk-reducing strategies for LS patients. Moreover, we present novel opportunities in the treatment and prevention of LS patients through harnessing the immune system using immunocheckpoint inhibitors and vaccines.

**Abstract:**

Lynch syndrome (LS) is the most common inherited disorder responsible for an increased risk of developing cancers at different sites, most frequently in the gastrointestinal and genitourinary tracts, caused by a germline pathogenic variant affecting the DNA mismatch repair system. Surveillance and risk-reducing procedures are currently available and warranted for LS patients, depending on underlying germline mutation, and are focused on relevant targets for early cancer diagnosis or primary prevention. Although pharmacological approaches for preventing LS-associated cancer development were started many years ago, to date, aspirin remains the most studied drug intervention and the only one suggested by the main surveillance guidelines, despite the conflicting findings. Furthermore, we also note that remarkable advances in anticancer drug discovery have given a significant boost to the application of novel immunological strategies such as immunocheckpoint inhibitors and vaccines, not only for cancer treatment, but also in a preventive setting. In this review, we outline the clinical, biologic, genetic, and morphological features of LS as well as the recent three-pathways carcinogenesis model. Furthermore, we provide an update on the dedicated screening, surveillance, and risk-reducing strategies for LS patients and describe emerging opportunities of harnessing the immune system.

## 1. Introduction

Lynch syndrome (LS), previously known as hereditary nonpolyposis colorectal cancer, is a genetic susceptibility to develop cancers at different sites, most commonly in the gastrointestinal and gynecological tracts, and is caused by a constitutive pathogenic variant responsible for mismatch repair (MMR) deficiency [1,2]. Importantly, LS patients can benefit from tailored cancer treatment, cancer surveillance, and prophylaxis including surgery and daily aspirin intake. 

Over the last decade, the role that diverse causative mutated genes have in the development of the disease has been better defined, leading to a diversification in the screening diagnostic path based on molecular alterations, and to a shift toward a more personalized health care approach. Although advances in our understanding have improved our ability to identify LS patients, LS still remains largely underdiagnosed, while timely identification is essential to guarantee appropriate medical care. 

In this review, we outline the clinical, biologic, genetic, and morphologic features specific of LS and display the recently proposed three-pathways model of carcinogenesis. In addition, we present the updated screening and surveillance strategies as well as the possible chemoprevention interventions for patients with LS, with a particular focus on novel immunological strategies.

## 2. Clinical Features and Diagnosis

LS is an inherited condition, with an estimated prevalence of up to 1 in 125 individuals in the general population, which predisposes to benign and malignant tumors [1,2]. Specifically, LS is the most common cause of both hereditary colon-rectal and endometrial carcinomas (ECs), accounting for 3–5% of colorectal carcinomas (CRCs) and for 2–6% of ECs [1,2,3]. Moreover, LS is associated with ovarian, gastric, small bowel, gallbladder, hepatobiliary tract, pancreas, urinary tract, kidney, brain, prostate cancers, and sebaceous skin tumors [4,5,6]. LS patients may manifest a Muir–Torre syndrome, which is the co-occurrence of a sebaceous skin tumor with any type of visceral cancer [1,7]. Some patients with LS develop many tumors at any age, while others do not develop tumors at all. LS-correlated cancers tend to have an earlier onset compared to sporadic carcinomas. Specifically, in LS patients, CRC occurs at an average age of 45–60 years (vs. 69 years) [4,8], EC at an average of 47–55 years (vs. 63 years) [2], and ovarian carcinoma at an average of 45 years (vs. 63 years) [9,10]. Cancer risk varies by sex, age, and mutation type [4,5,6].

Patients affected by constitutional mismatch repair deficiency syndrome (CMMRD), due to biallelic germline mutations, develop multiple intestinal adenomas at a very young age, unlike LS patients [11]. Moreover, CMMRD patients are predisposed to CRC, brain cancer, leukemia, lymphoma, skin lesions, and other DNA repair deficiency-related abnormalities. As a result, they may manifest a Turcot syndrome that is characterized by multiple colorectal adenomas together with brain cancer. 

Historically, before knowing the causal genes, the Amsterdam criteria were proposed, with the aim to identify patients likely to be affected by LS for research purposes (Table 1) [12]. Successively, the Bethesda guidelines were developed in order to select CRC patients to test for possible LS (Table 1) [13]. However, because of their low sensitivity, both the Amsterdam criteria and the Bethesda guidelines were later modified and updated to include patients with extracolonic cancers, later onset cancers, and no strong family history [14]. Nevertheless, these revised criteria still suffer from low sensitivity and, based on several studies, they have failed to identify more than one-fourth of LS patients, 20% of which will develop cancer over a 60-year lifespan [15]. This could result in potentially depriving patients of appropriate surveillance and timely therapy [16]. As a consequence, the current guidelines from the most important clinical associations recommend universal tumor screening using immunohistochemistry or microsatellite instability testing on all CRCs and ECs at the first diagnosis. This suggested approach gives the advantage of providing tailored treatment in patients with cancer and identifying relatives without cancer that can benefit from surveillance and prophylactic strategies [17,18].

## 3. Biological and Genetic Features

LS is an autosomal dominant genetic disorder due to a constitutional pathogenetic germline mutation in either the DNA MMR genes (i.e., *MLH1*, *MSH2*, *MSH6,* or *PMS2* genes), or *EPCAM* deletion. These mutations cause an MMR dysfunction that compromises the genomic stability and favor the accumulation of mutations at repetitive sequences of microsatellites [1,2]. 

In general, MMR genes are caretaker genes, since they oversee DNA MMR, and they contribute to maintaining the integrity and correctness of the sequence of DNA by correcting the misincorporation of nucleotides that may occasionally occur during replication. In humans, the heterodimers MutLα and hMutSα, composed of MLH1-PMS2 and MSH2-MSH6, respectively, mainly carry out the DNA MMR function (Figure 1). Importantly, MMR proficiency is indispensable to guarantee an accurate transmission of the genetic makeup, thus allowing a mutation rate of less than 2 × 10^−10^ base-substitutions per cell division [19]. 

A pathogenetic germline mutation in one of the four main MMR genes, or an *EPCAM* genomic deletion, predisposes to an MMR dysfunction, which in turn fully manifests itself when a second-hit random mutation affects the wild-type allele, causing a biallelic complete inactivation, consistent with Knudson’s two-hit model of carcinogenesis. Therefore, this second hit gives rise to cell MMR deficiency (MMRd), which is responsible for a mutator phenotype and results in microsatellite instability (MSI) [20]. As a consequence, there is an increasing mutation rate across the genome of somatic cells, which varies as a function of the affected gene and of the type of mutation. The buildup of spontaneous mutations predisposes to a high risk of cancer in LS-affected patients [21,22,23]. 

Microsatellites are short tandem repeat sequences of DNA, richly represented in the eukaryote genome, which appear every 50 kilobases throughout the genome. Since they are sensitive targets of DNA polymerase slippage during replication, they are subjected to deletion or insertion during replication, but are promptly corrected by a proficient MMR system. Conversely, in the case of MMRd, uncorrected errors result in numerous microsatellites of variable lengths, a phenotype also known as MSI. The MSI is typical of LS-correlated tumors. However, the degree of MSI is variable, as it depends on the specific MMR protein that is affected and on the mutation type. In particular, a MLH1 or MSH2 deficiency as well as a PMS2 loss lead to high-frequency MSI (MSI-H) of the mononucleotide repeats and short tandem repeats [20,24]. On the other hand, a deficit of MSH6 could also result in a low-degree MSI (MSI-L) that preferentially involves mononucleotide repeats [25].

Hence, the MSI may explain the key characteristic of LS-associated CRC, that is, faster carcinogenesis, as demonstrated by a rapid progression from a diminutive adenoma to a CRC in just 2–3 years as opposed to the 6–10-year timespan for sporadic CRC development [26]. Thus, MMRd cells, particularly those more actively proliferating, progressively accumulate random mutations that preferentially affect repetitive sequences, resulting in a frameshift mutation signature [27]. As a result, when a mutation falls in a region that interferes with the transcription or function of a cancer gene, it gives a selective advantage to the cell. Consequently, this transformed cell will then escape the normal proliferative and apoptotic control, leading to a clonal expansion with the accumulation of additional mutation. The chronological sequence of molecular events depends on the inherited germline mutation and on the degree of MMR functional impairment as well as on further acquired somatic mutations. Recently, Ahadova et al. [28] proposed a three-pathways carcinogenesis model in LS-affected patients that derives from CRC studies. This model outlines a stepwise molecular-morphological progression (Figure 2) [28,29,30,31].

Acting as tumor suppressor genes, DNA MMR genes are affected by inactivating germline mutations, mainly truncating mutations, which are frameshift mutations, nonsense mutations, large deletions, insertions, or inversions as well as splice site variants and missense mutations that modify the protein function. The exact type of mutation can vary among the different MMR genes [32]. Moreover, in a small proportion of cases, an epimutation, specifically the constitutional DNA methylation of the *MLH1* promoter, plays a role in mutation-negative LS-related cancers [33,34].

Notably, *EPCAM* does not belong to the MMR signaling pathway, but its genomic deletion in the 3′ end affects the downstream located *MSH2* gene, inducing *MSH2* promoter methylation with its consequent silencing in somatic cells and dysregulation of MMR.

Each gene gives a different contribution to the LS spectrum, depending on the specific role in the MMR machinery. Accordingly, *MLH1* and *MSH2,* which codify for the key proteins, are responsible for about 60–70% of the cases of LS, roughly evenly split, followed by *MSH6*, which accounts for 20–25%, *PMS2* estimated in less than 15% of cases, and *EPCAM* linked to just 1–3% of the cases [35,36]. Nevertheless, recent estimates report a substantially different prevalence of these germline mutations in the general population, with an unexpected higher proportion of mutations in *MSH6* and *PMS2* genes, likely underdiagnosed in LS families due to lower risk and lower penetrance of LS-related cancers [37]. Therefore, this apparent inconsistency depends on the definition, since defining LS based on the genotype also allows for the identification of patients with mute or partially expressed phenotype.

Importantly, the cumulative lifetime risk of cancer in LS carriers varies in the different organs as a function of gender and age, and also depends on the mutated gene and on the population (Figure 3) [4,5,6,38].

## 4. Pathological Features

In general, pathologic features of cancers with MSI/MMRd are similar in patients with germline mutations and somatic alterations, which are the majority and are caused by the somatic hypermethylation of the *MLH1* promoter. In particular, regardless of the cancer site, the status of MSI is often associated with an increased lymphocytic infiltration in the tumor, which is itself probably triggered by the high number of neoantigens displayed by the cancer cells (see below) [1,2,39]. In addition, carcinomas with MSI/MMRd tend to have a higher cytological grade, heterogeneous features, mucinous component, and dedifferentiation features [1,2]. Therefore, these histologic features may suggest the presence of MSI, but are not informative of the causative genetic basis.

However, a seminal study by Kloor et al. [40] reported the presence of MMRd crypt foci in the normal-appearing colonic mucosa of LS patients. Morphologically, these MMRd crypt foci were normal or had minimal morphological alterations, in the form of enlarged cell nuclei at the bottom of the crypt, crypt branching, and duplication. These foci were classified as monocryptic, oligocryptic, or policryptic. Initially, the MMRd crypt foci were estimated to be up to 1 per cm^2^ of colonic mucosa, but further studies were unable to confirm this high density and instead supported a lower density [41,42]. On the other hand, the number of MMRd crypt foci increased with age, in the left colon, and as a function of the specific variant of the MMR gene [41,42].

Analogously, three studies have demonstrated the presence of non-neoplastic endometrial glands with MMRd in endometrial samples from the MMR germline carriers [43,44,45]. These MMRd normal-appearing endometrial glands were found in up to 70% of LS patients, usually in large gland clusters and sometimes occupying most of the sampled endometrium [44,45]. Notably, both the MMRd crypt foci and the MMRd non-neoplastic endometrial glands are early putative precursors of LS-related cancers (Figure 2) and are both a specific hallmark of LS patients.

Many research studies on the immune landscape have focused on the differences between inherited and sporadic MMRd cancers, and sometimes resulted in inconsistent findings. In particular, most of these studies reported an increased T cell infiltration in LS-related MMRd CRCs compared to sporadic MMRd CRCs, even if others did not find significant differences between the two [46,47,48,49]. Furthermore, Takemoto et al. [48] reported a slightly increased number of stromal CD4- and CD8-positive T cells in the hereditary MMRd CRCs compared to the sporadic ones, while other authors failed to find significant differences [48,50,51].

Similarly, Pakish et al. [52] found increased stromal CD8-positive T cells and activated cytotoxic T cells together with reduced macrophages in LS-related MMRd ECs when compared to sporadic cases. However, Ramchander et al. [53] confirmed a significant increase in CD8-positive T cells only in the invasive margin of LS-associated MMRd ECs compared to the sporadic ones, but they could not find any increase in the tumor center. On the other hand, Sloan et al. [54] showed that the PD-L1 expression was higher and more frequent in the LS-related MMRd EC cells with respect to the sporadic cases, but another study found a significant association of PD-L1 positivity in the immune cell with *MLH1*-methylated ECs [55]. All of these data, showing a different immune profiling between inherited and sporadic MMRd cancers, support the role of the molecular mechanism underlying MMRd in shaping the immune landscape.

Notably, the cancer site in LS patients depends on the gender, age, affected mutation, familiar history, and geographical areas (Figure 3) [4,5]. Readers interested in more detailed epidemiologic and clinico-pathological characteristics of LS-related cancers by site can refer to previous comprehensive review articles [56,57].

## 5. Diagnostic Screening

In light of the weakness of family and clinical history (Amsterdam criteria and Bethesda guidelines) in recognizing LS families and the relevant contribution of LS to CRCs and ECs, universal screening of all of these cancers is suggested by the major medical associations in the field and is becoming a routine practice [17,18,58,59,60]. As delineated previously, more than 90% of LS-related cancers have high MSI due to MMRd, which can be easily investigated by immunohistochemistry. As a consequence, the recommended first-level screening tests for CRC are indiscriminately either MSI analysis by fluorescent polymerase chain reaction (PCR) or MMR proteins immunohistochemistry, since they have shown equivalent diagnostic performance in identifying MSI/MMRd cancers and similar cost-effectiveness, being both inexpensive methods [1,18]. Currently, the screening algorithm includes, in addition to MSI testing and MMR immunohistochemistry, *BRAF* mutation analysis and *MLH1* promoter methylation testing for CRCs, since MMRd sporadic cancers are mainly caused by *MLH1* promoter hypermethylation, and in the large bowel are associated with *BRAF*^V600E^ mutation (Figure 4) [1,18].

However, some authors have suggested immunohistochemistry over MSI testing, but others have recommended combining the two techniques in order to reduce the false negative rate and increase the sensitivity [16,61,62,63]. Specifically, in some studies, the MSI testing results were far less sensitive than MMR immunohistochemistry in ECs, therefore, the recommended test for EC is immunohistochemistry, followed by reflex *MLH1* promoter methylation testing in the case of the loss of MLH1 expression [63,64,65]. In the case of MSH2, MSH6, or isolated PMS2 protein loss or the absence of *MLH1* promoter hypermethylation, the patient should undertake a genetic testing of germline DNA.

These algorithms allow us to easily select cancer patients who should be referred to cancer genetic counseling and tested for MMR germline mutations, with the advantages of diagnosing LS patients with sentinel malignancy who may benefit from an individualized treatment plan and cancer-free relatives with LS who should start a specific surveillance program at a young age.

The rapid widespread of next-generation sequencing technologies, together with the progressive reduction in their price, make this technical approach appealing as an alternative to traditional syndrome specific genetic testing as well as in the up-front screening application. Currently, multigene panels are commercial and include several genes known or suspected to increase cancer risk including MMR genes. In the last decade, several studies have tested the use of multigene panel testing on unselected or high-risk patients [66,67]. In CRC and EC, multigene panel testing has proven the efficacy of current universal LS tumor screening strategies based on MMR immunohistochemistry and MSI testing, since they are able to detect the majority of LS-related cancers, with a sensitivity over 90% [66,68,69]. Regardless, it is conceivable that this technique will become the mainstream standard when the price becomes more competitive.

As supplementary methods, several clinical predictive models have been developed over the years to predict the probability of harboring an MMR germline mutation based on personal and family history [70,71,72,73]. These models including MMRpredict, MMRpro, PREMM_1,2,6_, and the most recent PREMM5 offer the major advantage among healthy subjects with a family history, and therefore, not testable with MMR immunohistochemistry and MSI in the absence of tumor, or patients with strong clinical suspicion but normal screening testing. In particular, PREMM5 analyzes the patients’ gender, age, personal and familial cancer history, and provides an accurate quantitative assessment of the individual risk of carrying a MMR germline mutation, instead of a binary classification. These models are cost-effective clinical tools to identify patients to refer to germline testing. In this way, they may support clinicians during the decision-making process and optimize the identification of LS carriers.

## 6. Immunology

The dysfunction of the MMR system causes the accumulation of frameshift mutations, resulting in the synthesis of many neoantigens that can act as not self-antigens or tumor-associated antigens [74]. In the complex immunosurveillance mechanisms against cancer, tumor-associated antigens can exert opposite roles. On one hand, they can be recognized as non-self and cleared through the elicited immune response. On the other hand, they can mask the epitopes on the cellular surface, eluding the antigen-presenting machinery and thereby facilitating an immune escape.

Frameshift peptides (FSPs) are able to induce both humoral- and cellular-mediated immune responses before the diagnosis of cancer. In fact, FSP-induced antibodies and FSP-specific effector T cell responses are detected in LS patients, also without cancer. In particular, FSP-specific effector T cell responses are commonly directed to neoantigens derived from genes with high mutation frequency, and with a so-called shared-landscape with MSI sporadic CRC [75].

Several findings suggest the relevance of the immune surveillance process in LS. The active surveillance exerted by the adaptive immunity could contribute to the reduced penetrance of the disease by controlling and suppressing the development of MMRd cancers. Such a hypothesis is corroborated by the increased growth of cancers observed in LS patients during or after treatment with immunosuppressant drugs [76]. In addition, the tumor microenvironment (TME) demonstrates a significant relevance in the local immune landscape of LS-related cancers because of its T cell infiltration and proinflammatory cytokine release [75].

However, many observed alterations could play an immunosuppressive role, among these is the antigen presentation process MCH class I mediated. A peculiar cause of fail in such machinery is due to a mutation in the beta2-microglobulin gene, reported in up to 30% of cases in MSI CRCs [77].

The advent of the so-called immunotherapy in oncology, in particular, antibodies against PD-1, PD-L1, and CTLA-4, have significantly modified the treatment, and consequently the history of many cancers, especially cancers with elevated tumor mutational burden, above all, neoplasms with MMRd/MSI [78,79]. Of note, the identification of shared cancer neoantigens in LS patients, able to evoke a targeted immune response, has been a cornerstone for developing vaccine-based therapies, useful not only in the treatment of LS-related cancers but also in a prevention setting.

## 7. Immunocheckpoint Inhibitors

Immunocheckpoint inhibitors (ICIs) are considered in this framework as a tissue/site agnostic therapy thus far. In 2017, the U.S. Food and Drug Administration granted accelerated approval to pembrolizumab for adult and pediatric patients with unresectable or metastatic MMRd/MSI-H solid tumors [80]. This approval was based on data from 149 patients with MMRd/MSI-H cancers enrolled across five uncontrolled, multi-cohort, multi-center, single-arm clinical trials.

Nowadays, ICIs are used in both metastatic sporadic and germline MMRd/MSI-H CRC and EC. A recent study investigated the efficacy to single-agent, PD-1 blockade by dostarlimab in patients with locally advanced MMRd rectal cancers [81]. All 12 patients that completed the treatment showed a clinical complete response, without severe adverse events or recurrence/progression during a minimum of 6 months follow-up.

Studies assessing the response of individuals with MMRd/MSI CRC to immune checkpoint blockade found no significant differences between the study participants with germline mutation and those with sporadic MMRd/MSI CRC receiving either monotherapy or combination therapy [12,82]. Even if some data are available from subgroup analyses, to date, there have been no large prospective studies specifically addressing the efficacy of ICIs in LS patients with CRC/EC.

## 8. Surveillance Strategies

Because of the high risk of developing CRC, a specific surveillance protocol must be offered to patients with LS. Many scientific societies have drawn up specific guidelines, defining both the type and timing of diagnostic procedures, and possibly risk-reducing colectomy on the basis of the colonoscopy results. The guidelines also indicate a specific surveillance protocol for uterus/ovary carcinomas (Table 2). As above-mentioned, the guidelines also recommend universal tumor screening for LS in all patients at the first diagnosis of CRC/EC, even if such a perfectly justified implementation could lead to a cascade of further investigations and procedures, along with potential challenges and difficulties [83].

In patients with clinical LS, only a single non-randomized controlled trial to date has investigated the efficacy of surveillance for CRC, reporting a reduction in the CRC rate and overall mortality by up to 62% and 65%, respectively, with colonoscopy at 3-year intervals over a 15-year period [84]. On the other hand, in almost 40% of LS patients, such a surveillance strategy could not prevent CRC development, likely because it was not preceded by a polyp and/or missed adenoma.

Since colonoscopic surveillance largely controls the CRC risk, the detection of extracolonic cancers becomes a top priority, particularly in LS women at the highest risk of EC. Nevertheless, no screening-trial results are hitherto available for extracolonic cancer. In general, recommendations for gynecological surveillance agree upon gynecological examination, transvaginal ultrasound, and endometrial biopsy, with or without CA125 dosage, starting at 30–35 years. Gastric surveillance with baseline endoscopy with the Helicobacter pylori test, and consequent eradication therapy if present, should also be indicated, mainly in regions and families with a high incidence of gastric cancer. Surveillance interventions in other organs should not be practiced due to the lack of evidence of effectiveness, except in the context of clinical trials or in high-risk patients, namely those with a family history of the specific cancers [85,86,87,88].

**Table 2 cancers-14-04102-t002:** The recent guidelines of surveillance strategy in Lynch syndrome patients.

Society Year	Colorectal Surveillance		Gynecological Surveillance			Other	Reference
Gene	Colonoscopy Interval	Age	Interval	Age	Procedures		
**AGA 2015**							
*MLH1*/*MSH2*	1–2 ys	20–25 ys *	NA	NA	NA	NA	[17]
*MSH6*/*PMS2*	1–2 ys	20–25 ys *					
**ASCO/ESMO 2015**							
*MLH1*/*MSH2*	1–2 ys	20–25 ys *	1 y	30–35 ys	Gynecological examination Pelvic US, Aspiration biopsy °	HP test and subsequent eradication for gastric cancer. Upper GI endoscopy every 1 to 3 ys in some high incidence populations	[88]
*MSH6*/*PMS2*	1–2 ys	20–25 ys *					
**ACG 2015**							
*MLH1*/*MSH2*	2 ys	20–25 ys	1 y	30–35 ys	TV US, endometrial biopsy °	Consider EGD with gastric biopsy and HP eradication, if found at 30–35 ys Gastric/duodenal cancer surveillance if family history	[87]
*MSH6*/*PMS2*	2 ys	20–25 ys					
**ESGE 2019**							
*MLH1*/*MSH2*	2 ys	25 ys	NA	NA	NA	Suggest HP test	[89]
*MSH6*/*PMS2*	2 ys	35 ys					
**ESMO 2019**							
*MLH1*/*MSH2*	1–2 ys	25 ys *	1 y	30–35 ys	TV US, CA125, End Biopsy	Consider gastric/ pancreatic/urinary tract cancer surveillance if family history	[18]
*MSH6*/*PMS2*	1–2 ys	35 ys *					
**BSG/ACPGBI/UKCGG 2020**							
*MLH1*/*MSH2*	2 ys	25 ys	NA	NA	NA	HP test and subsequent eradication for gastric cancer	[86]
*MSH6*/*PMS2*	2 ys	35 ys					
**EHTG/ESCG 2021**							
*MLH1*/*MSH2*	2–3 ys	25 ys	NA	NA	NA	No surveillance	[85]
*MSH6*	2–3 ys	35 ys					
*PMS2*	5 ys	35 ys					
**NCCN 2022**							
*MLH1*/*MSH2/* *EPCAM*	1–2 ys	20–25 ys ^+^	1–2 ys	30–35 ys	Consider Endometrial biopsy and TV US in postmenopause, serum CA125	Consider gastric/ pancreatic/urinary tract/ prostate/ skin cancer surveillance with risk factors	[90]
*MSH6*/*PMS2*	1–3 ys	30–35 ys ^+^					

* Or 5 years prior to youngest case in family; ^+^ Or 2–5 years prior to youngest case in family; ° Consider prophylactic hysterectomy with salpingo-oophorectomy when childbearing is completed; Acronyms: AGA—American Gastroenterological Association; ASCO—American Society of Clinical Oncology; ACG—American College of Gastroenterology; HP—Helicobacter pylori; ESGE—European Society of Gastrointestinal Endoscopy; ESMO—European Society for Medical Oncology; BSG—British Society of Gastroenterology; ACPGBI—Association of Coloproctology of Great Britain and Ireland; UKCGG—United Kingdom Cancer Genetics Group; NCCN—National Comprehensive Cancer Network; y—year; US—ultrasound; TV—transvaginal.

## 9. Prevention Interventions

### 9.1. Risk Reducing Surgery

No data from clinical trials are available regarding a survival advantage of extended colectomy in comparison with surveillance, so the latter represents the preferred option in most cases. Even if its role is controversial, a subtotal colectomy could be proposed to selected patients depending on the colonoscopy results. A recent systematic review with a meta-analysis about the comparison of selected versus extended colectomy showed that the latter reduced the risk of multiple CRCs by over four-fold compared with the segmental procedure [91].

The other risk-reducing surgery indicated in LS is a prophylactic hysterectomy, associated with bilateral salpingo-oophorectomy (HBSO), usually for women who have completed their childbearing or after menopause [92].

The Manchester International Consensus Group strongly recommends that HBSO is offered since 35–40 years of age, in heterozygotes with pathogenic *MLH1*, *MSH2*, and *MSH6*, whereas for pathogenic *PMS2* heterozygotes, there is not sufficient evidence. In this regard, a recent data analysis from a prospective LS database supports little benefit by performing HBSO before 40 years of age, and no measurable benefit for mortality in premenopausal carriers with pathogenetic *MSH6* or *PMS2* [93]. However, a general consensus does not exist because of the lack of strong data supporting such a recommendation [94].

### 9.2. Chemoprevention

Many studies have focused on efforts directed toward the chemoprevention of CRC in hereditary colon cancer syndromes, not only LS, but also familial adenomatous polyposis [95,96,97,98]. Numerous challenges occurred in such studies, the most important being the diverse gene-driven mechanisms responsible for the increased risk of cancer in different syndromes. Therefore, a targeted therapy must be tested only in people with a specific germline variant, with the obvious consequence of dramatically reducing the sample size of the studies.

Unfortunately, to date, no drug has received formal approval for this indication, even if aspirin represents the only pharmacological therapy specifically suggested for LS patients by some guidelines [90].

#### 9.2.1. Aspirin

The first findings regarding aspirin and its potential activity in reducing the incidence of CRC were reported in 1988, in the context of a case-control study [99]. In a recent review, Hybiak et al. [100] described the multiple effects of aspirin with a brief, but complete analysis of the studies regarding cancer chemoprevention with aspirin, mainly, but not only, CRC.

The most studied and plausible mechanism of action of aspirin as an anticancer is its effect as a cyclooxygenase (COX) inhibitor. In a dedicated review, Patrignani and Patrono provided an interesting framework, based on the phenotype switching of colonic cells toward cancer cells, a process induced by the platelets activated via the COX pathway [101]. Moreover, in a more recent paper, Serrano et al. [102] presented revised data and recommendations about aspirin and cancer chemoprevention in LS. At the moment, only one chemoprevention clinical trial in LS patients has been completed, the CAPP-2 (Cancer Prevention Program 2), and its results have been published. The first data compilation showed no effects on the incidence rate of CRC in more than 900 patients with clinical (Amsterdam criteria) or genetic diagnosis of LS treated with 600 mg of aspirin daily for up to 4 years in comparison with the placebo [103]. However, a further assessment at about 5 years of follow-up indicated a lower incidence of CRC in the intervention group, particularly in patients that had taken aspirin for almost 2 years [96]. Successively, this trend was confirmed and strengthened at the 10 year post-trial surveillance, which showed a reduced HR of 0.65 (95% CI 0.43–0.97; *p* = 0.035) for the aspirin versus placebo in the intention-to-treat group [104].

Actually, the main challenging task regards the optimal dose of aspirin in order to minimize the potential side effects without reducing the efficacy. To try to answer this, the CAPP-3 trial is currently ongoing, which aims to compare the effects of different doses of aspirin (100 mg, 300 mg, and 600 mg) in patients with LS. Analogously, at least two other ongoing trials have compared the efficacy of different doses of aspirin in CRC chemoprevention: the AAS-Lynch trial aims to evaluate the effect of 300 mg versus 100 mg versus the placebo for 4 years in LS patients [105]; and the ADD-aspirin protocol including four randomized phase 3 trial evaluating different doses of aspirin in patients over and under 75 years as an adjuvant therapy for different cancer, among these CRC, even if not specifically for LS patients [106].

#### 9.2.2. Non-Steroidal Anti-Inflammatory Drugs (NSAIDs) beyond Aspirin

Other studies that aimed to verify a protective effect of different NSAIDs on CRC development in LS were conducted. An observational study showed a reduced risk of CRC associated with aspirin and ibuprofen use. Based on their results, the authors estimated that the use of aspirin or ibuprofen for at least 1 month was associated with a reduced risk of CRC of about 60% and 65%, respectively [107].

Naproxen, a propionic acid derivative, has been evaluated in preclinical settings with promising results. The following naproxen trial was a Phase Ib study in which patients with LS were treated with HD naproxen, LD naproxen, or a placebo. The final analysis per protocol was performed on 54 patients and showed decreased mucosal prostaglandin E2 in the experimental arm, together with local activation of the resident immune cells at colonic mucosa, in a dose-dependent manner, but without evidence of a systemic recruitment or local proliferation of lymphocytes. Moreover, in a mouse co-clinical trial, the drug demonstrated a chemopreventive activity [98]. To note, naproxen seems to act through both canonical and non-canonical ways in the immune-modulation of the intestinal microenvironment. In particular, the low dose (220 mg) and high dose (440 mg) showed different effects in the downregulation and upregulation of different genes in the normal colonic mucosa, in both the stem cell and differentiated cell compartments [98].

Sulindac, a prodrug derived from sulfinylinden, has been studied because in preclinical data, it was shown to decrease the expression of cyclin B1 and E, and to induce the expression of a cell cycle inhibitor such as p21. However, in a randomized, double-blinded, placebo-controlled cross-over study in LS patients, the use of sulindac was associated with increased epithelial proliferation in the proximal colon. Because CRC in LS predominantly arises at this anatomical site, comprehensive concerns have been raised on the chemopreventive potential of sulindac casts [95].

#### 9.2.3. Progestins

Exogenous progestins are known to reduce the lifetime risk of EC and ovarian cancer in women in the general population, irrespective of MMR status [108,109,110]. Importantly, the protective effects of combined oral contraceptives become stronger the longer the use and persist for over 30 years [111,112]. In addition, the risk reduction for ovarian cancer is similar across different lifestyle factors, whereas for EC appears more pronounced in women with unhealthy lifestyle factors such as obesity, physical inactivity, and smoking [112].

In a phase II trial study, the use of depo-medroxyprogesterone acetate or combined oral contraceptives was specifically evaluated in patients with LS, showing evidence of a markedly decreased endometrial proliferation, which is considered as a surrogate of chemopreventive efficacy [113]. These data warrant further investigations on progestin agents in this high-risk population.

The levonorgestrel intrauterine system releases progestin directly into the endometrium, mitigating the stimulation effects of estrogens. Two studies showed that ever-users of the levonorgestrel intrauterine system have a reduced risk of EC and ovarian cancer [109,110]. Unfortunately, the POET, a randomized control phase III trial that intended to investigate the efficacy of levonorgestrel intrauterine system for the prevention of EC in women with LS, was prematurely closed due to poor recruitment (NTC00566644). On the other hand, a cohort retrospective study found that LS patients that received hormonal contraceptive for more than 1 year had a reduced risk of EC [114]. Hitherto, it still remains unknown whether the chemopreventive efficacy of hormonal contraceptives in LS patients is comparable to that of the general population.

### 9.3. Vaccines

The peculiar immunogenic properties of the majority of LS-associated polyps and cancers have led to hypothesizing a significant role of vaccines in reducing the cancer risk, especially in LS patients [115].

Particular interest was initially focused on the very high grade of immunogenicity exerted by both preinvasive and invasive LS-associated lesions. As above-mentioned, FSPs, also called neoantigens, are produced in the framework of MSI as a result of coding processes [116]. Given their ability to promote an immune response, FSPs have been considered an ideal target for vaccine-based therapies in LS with the aim of preventing malignant transformation.

A clinical phase I/II trial was performed to evaluate the safety and immunogenicity of an FSP-based vaccine (FSP neoantigens derived from mutant *AIM2, HT001, TAF1B* genes). While no safety concerns were reported, consistent induction of humoral and cellular immune responses was observed in all patients. In addition, one heavily pretreated patient with bulky metastases showed a stable disease with stable CEA levels over a period of 7 months [117].

Preclinical studies have reported that vaccination with a combination of four MSI-specific FSPs induces FSP-specific cellular and humoral responses in mice naïve or with *Msh2* conditional intestinal knockout. Additionally, in the latter mice closely simulating the clinical phenotype seen in LS-associated CRC patients, the tetravalent FSP vaccine reduced the intestinal tumor burden and prolonged overall survival. Furthermore, immune response, tumor growth reduction, and increased survival were all potentiated when daily naproxen was added [118].

In addition to neoantigens, tumor-associated antigens such as carcinoembryonic antigen (CEA) or mucin 1 (MUC1) have also demonstrated the ability to elicit an immune response. An ongoing randomized phase II trial is aimed at verifying the immunogenicity of a MUC1 peptide adjuvanted vaccine in patients with advanced adenomatous polyps (NCT02134925). These trial results will also be useful in understanding the potential role of such therapy in MMRd/MSI cancers, even when LS patients are not included.

Preliminary results from a phase I study with the Nous-209 vaccine, which contains 209 specific neoantigens, were presented to the ESMO Congress in 2021 [119]. This study, which administers the Nous-209 vaccine associated with pembrolizumab to patients with unresectable or metastatic MMRd/MSI-H CRC, gastric, or G-E junction cancers, is showing safety and highly immunogenic effects.

Furthermore, given the promising results of a preclinical study that combined the CEA-targeting vaccine with celecoxib for reducing intestinal polyps in an experimental model, further studies are desirable to test such association therapy [120,121]. In theory, a strong rationale also exists for the association of vaccines with other drugs such as NSAIDs, but also with immunotherapy or cytokines.

A comprehensive review on vaccines in LS has recently been published where the authors speculated about the potential use of liposomal RNA vaccines [76]. In fact, after their use during the COVID-19 pandemic, promising early data from a clinical trial on advanced melanoma were reported, which also prefigure a possible strong potential in the immunoprevention of other diseases such as LS.

### 9.4. Exercise Intervention and Other

A positive effect of the physical activity in reducing cancer risk can be derived from many studies [122,123]. Several hypotheses about the underlying mechanisms have been investigated such as increased antitumor immune response, decreased insulin and insulin-like growth factors, and possibly a reduced exposure of the mucosa to carcinogens following a reduced gastrointestinal transit time determined by exercise [124]. Similarly, the same mechanisms can be postulated to also play a role in reducing the cancer risk in LS patients.

Recent data from the Colon Cancer Family Registry have confirmed a potential beneficial effect of exercise intervention and reduced cancer risk in LS patients, but apparently only with a more intense activity compared to the standard recommendations (for instance, the guidelines from the United States Department of Health and Human Services recommend at least 2.5 h/week of moderate-intensity aerobic activity, and up to 5 h for more extensive health benefits) [125].

The potential role of resistant starch in reducing the risk of extracolonic cancer in LS patients has been reported. The CAPP2 trial separately studied the effects of both aspirin and resistant starch. Based on the results at 10 years, resistant starch exerts a protective effect against non-CRC, especially for cancers in the upper gastrointestinal tract [126]. Resistant starch, as a component of dietary fibers, could have an indirect effect on the immune system, a relevant property particularly for highly immunogenic neoplasms [127].

### 9.5. Ongoing Trials

In addition to the above-mentioned trials, other interesting studies are ongoing. One exploratory biomarker trial aims to assess the ability of atorvastatin alone or combined with aspirin to reduce the risk of CRC in high-risk individuals with LS (NCT04379999).

A single-arm, open-label study is currently underway to assess the effects of a moderate dose of omega-3-acid ethyl esters on the molecular and intestinal microbiota changes in participants at high-risk for CRC among the LS carriers (NCT03831698).

Additionally, mesalamine has been studied in chemoprevention for CRC. A first study, MesaCaPP4 (mesalamine for Colorectal Cancer Prevention program in Lynch Syndrome), for LS patients was terminated due to poor recruiting. A further multicenter, multinational, randomized, 2-arm, double-blind, phase II clinical study with the administration of 2000 mg mesalamine or placebo for a 2-year period in LS patients is ongoing. The aim of this study was to investigate the effect of regular treatment with mesalamine on the occurrence of any colorectal neoplasia, tumor multiplicity (the number of detected adenomas/carcinomas), and tumor progression in LS patients.

Of note, a phase III study is ongoing, aimed to explore the efficacy of tripleitriumab, a monoclonal antibody against PD-1, to prevent adenomatous polyps and second primary tumors in patients with LS (NCT04711434). Simultaneously, a phase II trial study (NCT03631641) is ongoing, aimed to determine how well nivolumab, another anti-PD-1 antibody, works in preventing colon adenomas in participants with LS and a history of segmental colon surgery.

Finally, a phase Ib/II trial aims to evaluate the safety and efficacy of the Nous-209 vaccine in LS patients (NCT05078866).

## 10. Conclusions

Recent evidence has helped us to better understand the role of different pathogenetic genes in the clinical manifestation and carcinogenesis of LS-related cancer, opening new perspectives for a more personalized cancer surveillance and prevention. In this landscape, old and emerging prevention intervention options such as pharmacologic therapies, immunotherapy, vaccines, and physical exercise can all give their contribution to a lower cancer risk of LS patients, along with a significant improvement in both the morbidity and mortality.

## Figures and Tables

**Figure 1 cancers-14-04102-f001:**
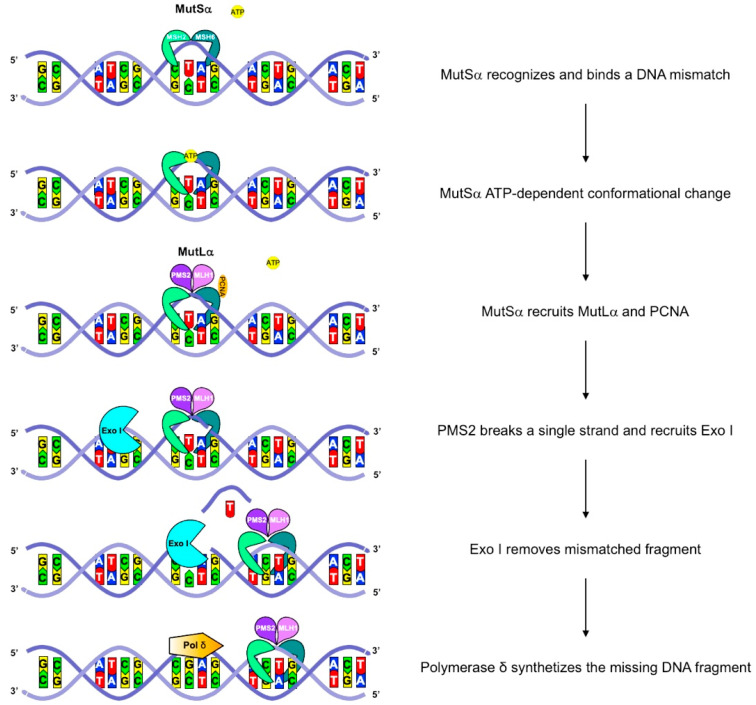
A schematic representation of the MMR machinery. In humans, the MMR machinery comprises nine conserved core proteins that are four MutL homologues, MLH1, MLH3, PMS1, and PMS2, and five MutS homologues, MSH2, MSH3, MSH4, MSH5, and MSH6. These proteins form hMutL heterodimers, composed of MLH1-PMS2 (hMutLα), MLH1-PMS1 (hMutLβ), and MLH1-MLH3 (hMutLγ). Both hMutLα and hMutLγ participate in insertion/deletion loop (IDL) repair, but single-base mismatches are mainly repaired by hMutLα. hMutS heterodimers constitute the mismatch-binding factors. Specifically, hMutSα is formed by MSH2 and MSH6 and binds single-base mispairs and IDLs, whereas hMutSβ is composed of MSH2 and MSH3 and mainly acts on IDLs. After binding, MutS undergoes an ATP-driven conformational change, allowing to recruit a hMutL heterodimer. The hMutS–hMutL complex recognizes a mismatched sequence, triggering the degradation of the nascently synthesized DNA by exonuclease I (exo I) and the correct sequence synthesis by polymerase δ.

**Figure 2 cancers-14-04102-f002:**
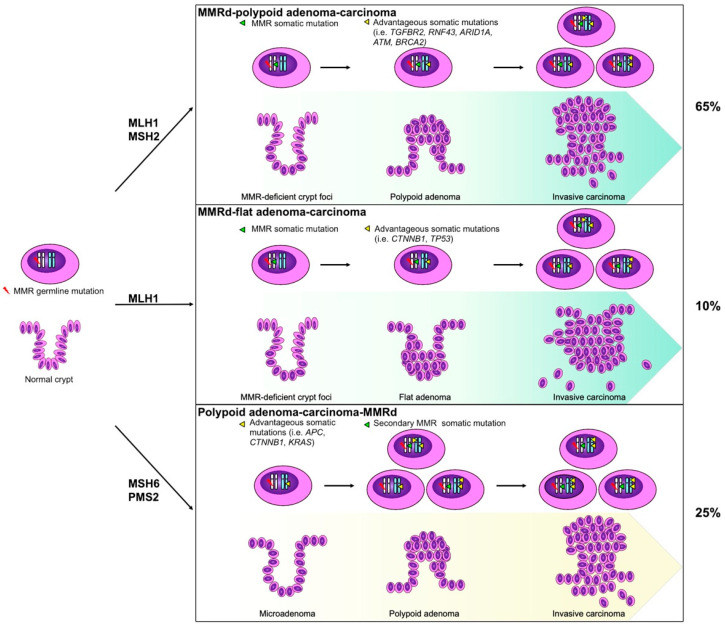
The three-pathways carcinogenic model in LS. A cartoon represents the hypothetical mechanisms of cancer development in LS with parallel intranuclear genetic events (above) and morphologic lesion in the colorectum (below). The somatic cell harboring a heterozygous germline mismatch repair (MMR) gene mutation is MMR-proficient and thus microsatellite stable (MSS). In the two upper models, the cells first acquire a somatic mutation in the same MMR gene, giving rise to MMRd crypt foci, morphologically normal, or only slightly altered, but with loss of MMR immunohistochemical staining. Then, depending on the further acquired mutation, they can progress to a polypoid lesion (with *TGFBR2*, *RNF43*, *ARID1A*, *ATM,* or *BRCA* mutation) or a flat lesion (in case of *CTNNB1* or *TP53* mutation), and thus becomes invasive. The third model is analogous to the sporadic microsatellite stable process, and MMRd can occur as a late event.

**Figure 3 cancers-14-04102-f003:**
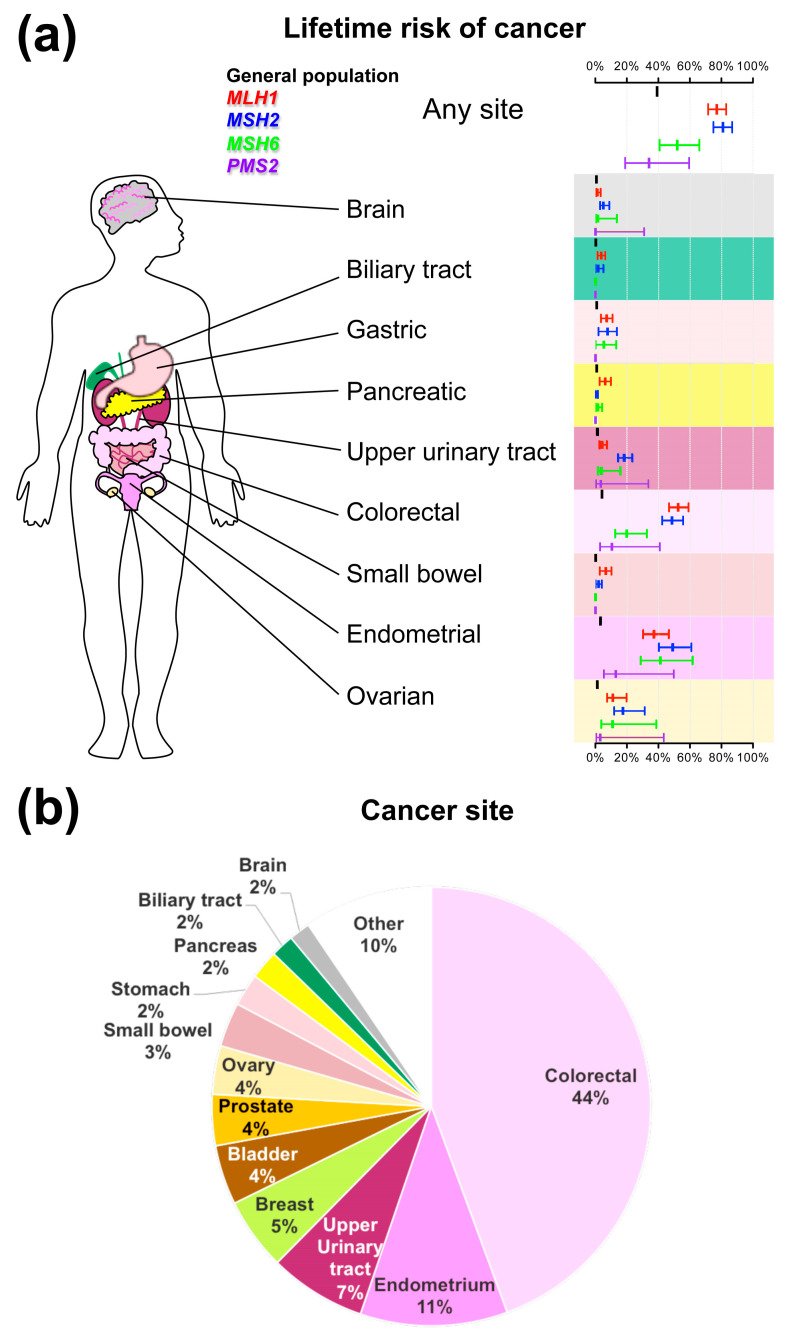
(**a**) The lifetime risk of cancer in patients with Lynch syndrome based on pathogenic MMR gene and site. The risks of endometrial and ovarian cancer were calculated in a female population. (**b**) Frequency of cancer by site in 6350 carriers of pathogenic MMR variants from the Prospective Lynch Syndrome Database (skin cancer were excluded) [5].

**Figure 4 cancers-14-04102-f004:**
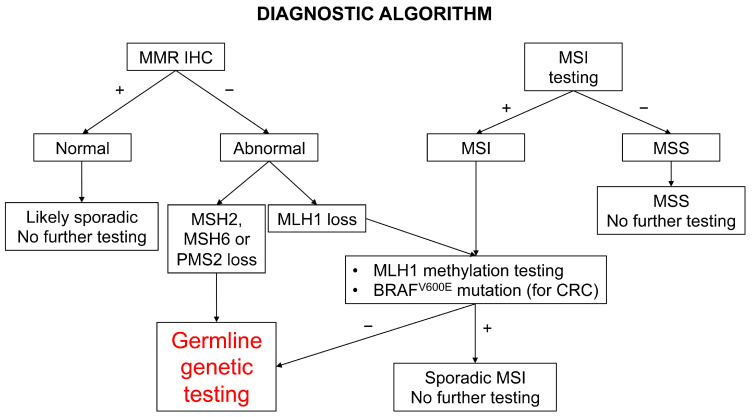
The diagnostic algorithm for colorectal (CRC) and endometrial cancer. MMR deficiency can be tested with immunohistochemistry (IHC) for MLH1, MSH2, MSH6, and PMS2 or MSI testing, complemented by *MLH1* promoter methylation testing, and *BRAF* mutation analysis for CRCs.

**Table 1 cancers-14-04102-t001:** Revised Amsterdam II criteria and Bethesda guidelines for the clinical diagnosis of Lynch syndrome.

Amsterdam II Revised Criteria
All of the following criteria should be met:
Three or more blood relatives with a Lynch-related cancer (CRC, endometrial, small bowel, ureter, or renal pelvis)
One relative must be a first-degree relative of the other two
One or more cancer cases diagnosed before the age of 50 years
Two or more successive generations affected
Familial adenomatous polyposis excluded in CRC
Tumor diagnosis confirmed by histopathologic examination
**Revised Bethesda Guidelines**
At least one of the following criteria should be met:
CRC or endometrial cancer diagnosed before the age if 50 years
Synchronous or metachronous CRC or other LS-associated tumors, regardless of age
Colorectal cancer with MSI-high-associated morphologic features (Crohn-like lymphocytic reaction, mu-cinous/signet cell differentiation, or medullary growth pattern) in a patient younger than 60 years
CRC in a patient with one or more first-degree relatives with an LS-associated tumor, with one of the can-cers being diagnosed before the age of 50 years
CRC diagnosed in two or more first- or second-degree relatives with LS-associated tumors, regardless of age

CRC—colorectal cancer; LS—Lynch syndrome; MSI—microsatellite instability.

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
