# Peer review of "Lynch Syndrome: From Carcinogenesis to Prevention Interventions"

_cancers, 2022, doi:10.3390/cancers14174102_

Round 1

Reviewer 1 Report

Gambini et al., provide a comprehensive review of the literature primarily focusing on emerging clinical practice concerning treatment of Lynch syndrome (LS) patients, including prevention strategies. All aspects of LS, starting from its diagnosis to treatment and prevention, have experienced significant shifts in established clinical practice in recent times. My following suggestions are directed towards improving readability and making the review more informative. 

  1. - Line # 63-64 and 66-67 need appropriate references.  

  1. - I find the paragraph from line # 68-75 to be digressive and may be removed completely. 

  1. - Instead, authors could include a description regarding the average age of presentation of LS patients in comparison to normal population. This information is critical as it is a crucial piece of information for suspecting LS and is currently missing from the review. This will be further re-emphasized in the diagnostic algorithm described later in the text. 

  1. - Line # 102. It is better to state that mismatch repair (MMR) corrects occasional misincorporation of nucleotides by DNA replicating polymerases. Kindly consider rephrasing. 

  1. - Fig. 1 legend. Line # 119. Instead of using “wrong fragment”, it is more appropriate to use “nascently synthesized DNA”. 

  1. - Line# 125-126. Mutator phenotype is generally described to indicate deficient MMR condition which results in/manifests as MSI. The current description in the text seems to indicate the other way around. Kindly reconsider. 

  1. - Fig. 2 legend. Line # 160. I think the authors meant to write “haplo-sufficient”, and not “haplo-insufficient”. Kindly check. 

  1. - Line # 263. There is a typo between “MSH6 e PMS2”. 

  1. - Line # 325. Typo in beta2mycroglobuline gene. 

  1. - Under “immune-checkpoint inhibitors (ICI)”, maybe it is worth mentioning the recent NEJM study where complete clinical remission was achieved with ICI as a single agent in locally advanced rectal cancer. (PMID: 35660797) 

  1. - Authors have duly discussed NCCN guidelines of 2021. However, I urge the authors to refer to NCCN guidelines 2.2022 which was recently released and incorporate any significant changes to make the review more up to date. 

  1. - In a similar vein, maybe it is worth discussing the recent publication of resistant starch supplementation benefits for LS patients. (Mathers et al., 2022, PMID: 35878732) 

Author Response

We thank the reviewer and report our point-by-point replies to her/his comments

-Line # 63-64 and 66-67 need appropriate references.

Answer: As suggested, we added appropriate references (references 4-6). 

- I find the paragraph from line # 68-75 to be digressive and may be removed completely. 

Answer: As suggested, we synthesized this paragraph (line # 74-80).

-Instead, authors could include a description regarding the average age of presentation of LS patients in comparison to normal population. This information is critical as it is a crucial piece of information for suspecting LS and is currently missing from the review. This will be further re-emphasized in the diagnostic algorithm described later in the text. 

Answer: we reported the average age of presentation of the principal LS-related cancers (line # 69-73)

- Line # 102. It is better to state that mismatch repair (MMR) corrects occasional misincorporation of nucleotides by DNA replicating polymerases. Kindly consider rephrasing. 

Answer: we rephrased as suggested (line # 108)

- Fig. 1 legend. Line # 119. Instead of using “wrong fragment”, it is more appropriate to use “nascently synthesized DNA”. 

Answer: we corrected as suggested (line # 125).

- Line# 125-126. Mutator phenotype is generally described to indicate deficient MMR condition which results in/manifests as MSI. The current description in the text seems to indicate the other way around. Kindly reconsider. 

Answer: we rephrased as suggested (line # 131)

- Fig. 2 legend. Line # 160. I think the authors meant to write “haplo-sufficient”, and not “haplo-insufficient”. Kindly check.

Answer: we deleted the haploinsufficient (line # 168).

-Line # 263. There is a typo between “MSH6 e PMS2”. 

Answer: we corrected to “MSH6 and PMS2” (line # 271)

- Line # 325. Typo in beta2mycroglobuline gene. 

Answer: we corrected the typo (line # 333)

- Under “immune-checkpoint inhibitors (ICI)”, maybe it is worth mentioning the recent NEJM study where complete clinical remission was achieved with ICI as a single agent in locally advanced rectal cancer. (PMID: 35660797) 

Answer: we thank the reviewer for the insightful suggestion. We mentioned the suggested study (line # 350-354)

- Authors have duly discussed NCCN guidelines of 2021. However, I urge the authors to refer to NCCN guidelines 2.2022 which was recently released and incorporate any significant changes to make the review more up to date. 

Answer: we updated the reference incorporating the significant changes (Table 2, reference 90).

- In a similar vein, maybe it is worth discussing the recent publication of resistant starch supplementation benefits for LS patients. (Mathers et al., 2022, PMID: 35878732) 

Answer: we discussed the suggested reference (line # 569-575).

Reviewer 2 Report

This review article aims to provide an overview of Lynch Syndrome from several perspectives. Beginning with clinical features, it then focuses on the molecular biology and genetics, and expands to tumour characteristics, diagnostics, immunology, surveillance (including tabulation of guidelines), and finally  preventions.  This is a well-written, broad and detailed review article. It is up-to-date with recent publications.  It covers many aspects of Lynch Syndrome, so will appeal to readers who want to expand their understanding on any particular area, noting that a diverse range of expertise can be informed from this review. The article is well-balanced, and provides a suitable examination of each aspect of Lynch Syndrome with in-depth analysis. 

Some minor grammar suggestions:

line 266: "techniques" instead of technics

line  267: "MSI testing results are far less sensitive...."

line 321: "because of its T cell infiltrate and proinflammatory" -> T cell infiltration  

Author Response

We thank the reviewer and corrected the manuscript as suggested by the reviewer.